# Emergence and Spread of a B.1.1.28-Derived P.6 Lineage with Q675H and Q677H Spike Mutations in Uruguay

**DOI:** 10.3390/v13091801

**Published:** 2021-09-10

**Authors:** Natalia Rego, Cecilia Salazar, Mercedes Paz, Alicia Costábile, Alvaro Fajardo, Ignacio Ferrés, Paula Perbolianachis, Tamara Fernández-Calero, Veronica Noya, Matias R. Machado, Mariana Brandes, Rodrigo Arce, Mailen Arleo, Tania Possi, Natalia Reyes, María Noel Bentancor, Andrés Lizasoain, Viviana Bortagaray, Ana Moller, Odhille Chappos, Nicolas Nin, Javier Hurtado, Melissa Duquía, Maria Belén González, Luciana Griffero, Mauricio Méndez, Maria Pía Techera, Juan Zanetti, Emiliano Pereira, Bernardina Rivera, Matías Maidana, Martina Alonso, Pablo Smircich, Ighor Arantes, Daiana Mir, Cecilia Alonso, Julio Medina, Henry Albornoz, Rodney Colina, Gonzalo Bello, Pilar Moreno, Gonzalo Moratorio, Gregorio Iraola, Lucía Spangenberg

**Affiliations:** 1Bioinformatics Unit, Institut Pasteur de Montevideo, Montevideo 11400, Uruguay; natalia@pasteur.edu.uy (N.R.); tamfer@pasteur.edu.uy (T.F.-C.); mbrandes@pasteur.edu.uy (M.B.); 2Laboratorio de Genómica Microbiana, Institut Pasteur Montevideo, Montevideo 11400, Uruguay; csalazar@pasteur.edu.uy (C.S.); iferres@pasteur.edu.uy (I.F.); 3Centro de Innovación en Vigilancia Epidemiológica, Institut Pasteur Montevideo, Montevideo 11400, Uruguay; mpaz@pasteur.edu.uy (M.P.); acostabile@fcien.edu.uy (A.C.); 4Laboratorio de Virología Molecular, Facultad de Ciencias, Universidad de la República, Montevideo 11400, Uruguay; afajardo@fcien.edu.uy (A.F.); paulaperbolianachis@fcien.edu.uy (P.P.); arcerama@gmail.com (R.A.); 5Laboratorio de Evolución Experimental de Virus, Institut Pasteur de Montevideo, Montevideo 11400, Uruguay; 6Sección Bioquímica, Facultad de Ciencias, Universidad de la República, Montevideo 11400, Uruguay; 7Department of Exact and Natural Sciences, Universidad Católica del Uruguay, Montevideo 11600, Uruguay; 8Laboratorio de Biología Molecular, Sanatorio Americano, Montevideo 11600, Uruguay; veronicanoyaro@gmail.com (V.N.); mailenarleo@gmail.com (M.A.); tpossi@pasteur.edu.uy (T.P.); reyesntl@gmail.com (N.R.); manoben2011@gmail.com (M.N.B.); 9Protein Engineering, Institut Pasteur de Montevideo, Montevideo 11400, Uruguay; mmachado@pasteur.edu.uy; 10Laboratorio de Virología Molecular, Departamento de Ciencias Biológicas, CENUR Litoral Norte, Universidad de la República, Salto 50000, Uruguay; andres.lizasoain.cuelho@gmail.com (A.L.); viviborta@gmail.com (V.B.); anaclaramoller2@gmail.com (A.M.); rodneycolina1@gmail.com (R.C.); 11Centro Universitario Regional Este, Universidad de la República, Rocha 27000, Uruguay; izisodhille@gmail.com (O.C.); meliduquia@gmail.com (M.D.); mabel2406@gmail.com (M.B.G.); luciana.griffero@cure.edu.uy (L.G.); mauriciomendezpeyre@gmail.com (M.M.); mariapiatechera@gmail.com (M.P.T.); juanzanetti04@hotmail.com (J.Z.); pereiramemo@gmail.com (E.P.); ceci.babilonia@gmail.com (C.A.); 12Unidad de Cuidados Intensivos, Hospital Español “Juan J. Crottogini”, Montevideo 11800, Uruguay; niconin@hotmail.com (N.N.); javierhurtado2005@gmail.com (J.H.); 13Laboratorio de Diagnóstico Molecular, Institut Pasteur de Montevideo, Montevideo 11400, Uruguay; bernardinariverasoto@gmail.com (B.R.); maidana@pasteur.edu.uy (M.M.); martina.alonso60@gmail.com (M.A.); 14Bioinformatics Laboratory, Department of Genomics, Instituto de Investigaciones Biológicas Clemente Estable, MEC, Montevideo 11600, Uruguay; psmircich@gmail.com; 15Laboratory of Molecular Interactions, Facultad de Ciencias, UdelaR, Montevideo 11400, Uruguay; 16Laboratorio de AIDS e Imunologia Molecular, Instituto Oswaldo Cruz, Fiocruz, Rio de Janeiro 21040-900, Brazil; ighorarantes@gmail.com (I.A.); gbellobr@gmail.com (G.B.); 17Unidad de Genómica y Bioinformática, Departamento de Ciencias Biológicas, Centro Universitario Regional Litoral Norte, Universidad de la República, Salto 50000, Uruguay; daianamir@gmail.com; 18Cátedra de Enfermedades Infecciosas, Facultad de Medicina, Universidad de la República, Montevideo 11300, Uruguay; jcmedina1@gmail.com (J.M.); henry_albornoz@hotmail.com (H.A.); 19Dirección General de Salud, Ministerio de Salud Pública, Montevideo 11200, Uruguay; 20Center for Integrative Biology, Universidad Mayor, Santiago de Chile 8580745, Chile; 21Host-Microbiota Interactions Laboratory, Wellcome Sanger Institute, Hinxton, Cambridge CB10 1SA, UK; 22Department of Informatics and Computer Science, Universidad Católica del Uruguay, Montevideo 11600, Uruguay

**Keywords:** SARS-CoV-2, B.1.1.28, Spike, Q675H, Q677H, Uruguay, phylogeography, phylogenetics

## Abstract

Uruguay controlled the viral dissemination during the first nine months of the SARS-CoV-2 pandemic. Unfortunately, towards the end of 2020, the number of daily new cases exponentially increased. Herein, we analyzed the country-wide genetic diversity of SARS-CoV-2 between November 2020 and April 2021. We identified that the most prevalent viral variant during the first epidemic wave in Uruguay (December 2020–February 2021) was a B.1.1.28 sublineage carrying Spike mutations Q675H + Q677H, now designated as P.6, followed by lineages P.2 and P.7. P.6 probably arose around November 2020, in Montevideo, Uruguay’s capital department, and rapidly spread to other departments, with evidence of further local transmission clusters; it also spread sporadically to the USA and Spain. The more efficient dissemination of lineage P.6 with respect to P.2 and P.7 and the presence of mutations (Q675H and Q677H) in the proximity of the key cleavage site at the S1/S2 boundary suggest that P.6 may be more transmissible than other lineages co-circulating in Uruguay. Although P.6 was replaced by the variant of concern (VOC) P.1 as the predominant lineage in Uruguay since April 2021, the monitoring of the concurrent emergence of Q675H + Q677H in VOCs should be of worldwide interest.

## 1. Introduction

By the end of 2020 and the beginning of 2021, several studies reported the emergence of novel SARS-CoV-2 variants of interest (VOIs) and concern (VOCs) with different missense mutations and deletions in the Spike (S) protein that impact viral transmissibility and escape from previous host’s immune responses, among other features [1,2]. In Brazil, the SARS-CoV-2 lineages B.1.1.28 and B.1.1.33 dominated the first epidemic wave [3,4], but were replaced by VOC P.1 (WHO name: Gamma) and P.2 (former VOI Zeta), both descendants of lineage B.1.1.28, by the end of 2020 and beginning of 2021 [5]. So far, five B.1.1.28 descendant sublineages carrying mutations of concern have emerged. The VOC P.1, which harbors the mutations of concern S:K417T/E484K/N501Y among its lineage defining mutations [6], originated in the Amazonas state in mid-November [7,8] and rapidly spread across Brazil and to over 50 countries globally [9]. The lineages P.2, P.4, and P.5, carrying the concerning amino acid changes S:E484K, S:L452R, and S:E484Q/N501T, respectively, were also initially detected in samples from Brazil [10,11,12,13,14,15]. The lineage P.3 (former VOI Theta) emerged in the Philippines, and it includes substitutions S:E484K/N501Y/P681H among the lineage-defining mutations; the first sample was collected on 8 January 2021, and later it further spread to the USA, Germany, and Malaysia, among other countries [16]. One additional B.1.1.28 descendant clade that emerged in southern Brazil carrying mutation N:P13L was recently defined as a new Pango P.7 lineage [17,18].

Uruguay was able to control the early viral dissemination during the first nine months of the SARS-CoV-2 pandemic by implementing a successful Test, Trace, and Isolation strategy (TETRIS). The low number of total cases, contained outbreaks, and few deaths were characteristic for this first period [19,20]. At the beginning, viral diversity was high, with cocirculation of strains A.2, A.5, B.1, B.1.195, and B.31, introduced mostly through Montevideo, Uruguay’s capital city and connection hub through its international airport and harbor [21]. Later, multiple introductions of SARS-CoV-2 lineages B.1.1.28 and B.1.1.33 of Brazilian origin were detected in Uruguay, mainly along the 1068 km long Uruguayan–Brazilian dry border, and these lineages became predominant between May and July 2020 [22]. Towards the end of 2020, the number of active cases exponentially increased, from an average of 60 cases per day during October and November to more than 400 during December [19], concomitant with the loss of the TETRIS safety zone [23,24]. SARS-CoV-2 positive daily new cases decreased around mid-February after more stringent mobility measures were taken by the government [20,25]; but the total number of cases stayed outside the TETRIS zone and a second exponential growth period started in March 2021, coinciding with the introduction and dissemination of VOC P.1 [26]. Summer-related social gatherings and relaxed social distancing are some of the proposed reasons to explain the first epidemic wave in Uruguay, but there is currently a gap in knowledge concerning the potential influence of virological factors. 

To understand the SARS-CoV-2 diversity associated with the first COVID-19 epidemic wave in Uruguay, we conducted a retrospective epidemiological and genomic analysis of SARS-CoV-2 complete genomes from COVID-19 patients diagnosed between November 2020 and April 2021. Our study revealed that a novel B.1.1.28 clade harboring two nonsynonymous changes in the Spike protein: Q675H and Q677H, now designated as lineage P.6 [27], was the most prevalent SARS-CoV-2 variant by the end of 2020 and beginning of 2021. Lineages P.2 and P.7 were also detected at lower prevalence during the first epidemic wave in Uruguay. The Q675H and Q677H mutations are in the proximity of the polybasic cleavage site at the S1/S2 boundary, a region of biological relevance for virus replication [28], and also arose independently in many other SARS-CoV-2 VOIs circulating worldwide. These findings suggest that local emergence and spread of a more transmissible P.6 variant might have had a non-negligible role in the first epidemic wave of COVID-19 in Uruguay.

## 2. Materials and Methods

### 2.1. Ethics Statement

This work was done by the Inter-Institutional Working Group (IiWG) for SARS-CoV-2 genomic surveillance in Uruguay, which involves a diagnostic network, expertise and resources to handle large-scale sequencing, computational scientists for genomic analysis, and an affordable and decentralized “in-house” qPCR test designed to detect known VOCs [26]. Residual deidentified RNA samples from SARS-CoV-2 positive patients were remitted to the Institut Pasteur de Montevideo (IPMon). IPMon was validated by the Ministry of Health of Uruguay as an approved center providing diagnostic testing for COVID-19. All samples were deidentified before receipt by the study investigators. All relevant ethical guidelines were appropriately followed. Additionally, the project was approved by the Ethics Committee of the Sanatorio Americano SASA (Uruguay) on the 29 April 2021. Ethical approval was given, and signed informed consent was obtained from the participants.

### 2.2. SARS-CoV-2 Samples

In total, 260 SARS-CoV-2 RNA samples (Appendix A) were recovered from nasopharyngeal–throat combined swabs collected from clinically ill or asymptomatic individuals that resided in different Uruguayan departments and were diagnosed from November 2020 to April 2021 in Uruguay. As the IiWG began working in March 2021, the availability of earlier samples was conditional to what was kept at any laboratory of the IiWG diagnostic network. Positive RNA samples were reverse transcribed using SuperScript™ II Reverse Transcriptase (Thermo Fisher Scientific Inc., Waltham, MA, USA) or the LunaScript^®^ RT SuperMix Kit (New England Biolabs, Ipswich, MA, USA). A negative control was included at this point and carried throughout the protocol. 

### 2.3. Genome Sequencing

Sequencing libraries were prepared according to the classic ARTIC protocol described by Quick J. [29,30], the 2000 bp long amplicon version described by Resende P.C. [31], or the sequencing protocol using a 1200 bp amplicon “midnight” primer set, with the Nanopore Rapid kit as described by Freed N. and Silander O. [32,33] (Appendix A). The final library was eluted in EB buffer (ONT) and quantified using a fluorometric assay. Recommended amounts of library were loaded into a FLO-MIN106D R9.4.1 flowcell and sequenced on the MinION Mk1C or GridION X5 sequencing platforms (ONT). Basecalling and demultiplexing were performed with Guppy 4.3.2 or higher [34] using the high or super accuracy mode. Consensus genomes were generated using the poreCov pipeline 0.7.0 or higher [35,36,37,38,39,40,41,42,43,44], and Nanopolish was used for consensus generation. Complete sequences with up to 15% of Ns were kept for further analysis. All genomes obtained in this study were uploaded to the EpiCoV database in the GISAID initiative under the accession numbers specified in Appendix A.

### 2.4. SARS-CoV-2 Lineage Assignment

SARS-CoV-2 full-length consensus sequences were manually curated in specific genome positions, such as clade-defining mutations. Genotyping was performed according to Rambaut et al. [45] using the Pangolin application [46,47], and later confirmed using maximum likelihood (ML) analysis.

### 2.5. Phylogenetic and Phylogeographic Analysis of B.1.1.28

Uruguayan B.1.1.28 sequences (*n* = 212; samples assigned as lineage B.1.1.28, P.6, or P.7 in Appendix A) were next analyzed in the context of additional B.1.1.28 sequences from Uruguay and Brazil, downloaded from the EpiCoV database of the GISAID initiative [48] (Appendix A showing EpiCoV/GISAID acknowledgments). Downloaded B.1.1.28 sequences from Uruguay (*n* = 143) were complete, with full collection date information and sampled before 31 May 2021. Sequences from Brazil (*n* = 1428) were complete and high quality with full collection date information, and were sampled before 31 May 2021 (Appendix A). Additionally, we downloaded four B.1.1.28 from the USA (*n* = 2), Spain (*n* = 1), and Belgium (*n* = 1) that also harbored both S:Q675H and S:Q677H mutations (Appendix A). Alignment was performed with MAFFT v7.471 [49]. Maximum likelihood phylogenetic analysis of the 1787 B.1.1.28 sequences was performed with IQ-TREE version 1.6.12 under the model GTR + F + R3 of nucleotide substitution selected by the built-in ModelFinder option [50]. Branch support was assessed by the approximate likelihood-ratio test based on a Shimodaira–Hasegawa-like procedure (SH-aLRT) with 1000 replicates [51]. The tree root was established with the sequence EPI_ISL_416036 with the earliest collection date of 5 March 2020. This tree was time-scaled using TreeTime 0.8.3.1 [52], applying a fixed clock rate of 8 × 10^−4^ substitutions/site/year [53,54], and keeping polytomies. The time-scaled tree was then employed for the ancestral character state reconstruction (ACR) of epidemic locations with PastML v.1.9.15 [55], using the marginal posterior probabilities approximation (MPPA) method with an F81-like model. Brazilian sequences were grouped according to the region: South, Southeast, Central West, North, and Northeast. A time-scaled Bayesian phylogeographic analysis was next performed to infer the geographical source and dissemination pattern of the Uruguayan B.1.1.28 + Q675H + Q677H (now P.6) samples, and to estimate the time of their most recent common ancestors (T_MRCA_). Phylogenetic trees were estimated in BEAST v1.10 [56] using the GTR + F + I nucleotide substitution model, the nonparametric Bayesian skyline model as the coalescent tree prior [57], a strict molecular clock model with a uniform substitution rate prior (8–10 × 10^−4^ substitutions/site/year), and a reversible discrete phylogeographic model (using Uruguayan departments as epidemic locations) [58] with a continuous-time Markov chain (CTMC) rate reference prior [59]. MCMC chains were run for 100 million generations, and convergence (effective sample size > 200) in parameter estimates was assessed using Tracer v1.7 [60]. The maximum clade credibility (MCC) tree was summarized with TreeAnnotator v1.10 [61] and visualized using FigTree v1.4.4 [62]. Additional visualizations were implemented in the R environment with treeio 1.16.2 and ggtree 3.0.4 Bioconductor packages [63].

### 2.6. Phylogenetic and Phylogeographic Analysis of P.2

Uruguayan P.2 sequences (*n* = 79) were analyzed in the context of 1272 additional Brazilian P.2 sequences downloaded from EpiCoV/GISAID; sequences were complete and high quality with full collection date information, and were sampled before 31 May 2021 (Appendix A). Five Brazilian sequences with a collection date before September 2020 were discarded (the collection date might have been incorrect). Alignment and ML phylogenetic analysis were performed as above. The tree root was established with the sequence EPI_ISL_2344425 with the earliest collection date of 1 September 2020. The tree was time-scaled using TreeTime [52] and the ACR of epidemic locations inferred with PastML [55], as before. Brazilian locations were grouped according to the five Brazilian regions mentioned above. The resulting tree with inferred locations was visualized using FigTree [62].

### 2.7. Lineage Prevalence of Available Uruguayan Samples

To assess the prevalence of B.1.1.28 + Q675H + Q677H clade in the context of the different SARS-CoV-2 lineages circulating in Uruguay, we used the lineage assignments obtained for the 260 samples presented in this study, 335 additional samples already available at EpiCoV/GISAID, and 342 records available at the IiWG domain. As only one B.1.1.28 (not carrying Q675H + Q677H mutations) was identified in May 2021, we kept our analysis from November 2020 to April 2021. The dataset comprised a total of 937 SARS-CoV-2 genotypes, and prevalence (relative frequency) of each lineage was calculated monthly, from November to April. The “Others” category included B.1, B.1.1, B.1.1.1, B.1.1.34, B.1.177, B.1.177.12, and B.1.238. 

In the case of geographic distribution of P.6 and P.2 variants (for the Uruguayan maps), the number of cases per department were obtained using samples from this study and also available at the IiWG domain, from November 2020 to April 2021 (*n* = 174 for P.6 and *n* = 70 for P.2).

### 2.8. Determination of Prevalence of Q675H + Q677H

To assess the prevalence of co-occurring S:Q675H and S:Q677H in worldwide SARS-CoV-2 genomes, we downloaded from EpiCoV/GISAID (accessed on 7 July 2021) 129 complete genomes, with high quality and full collection date information. We removed redundant samples with the ones included in this study, obtaining a final dataset of 259 sequences (Appendix A).

### 2.9. Structural Representation of the Spike Protein

The molecular model of SARS-CoV-2 Spike glycoprotein was taken from the D. E. Shaw Research database (DESRES-ANTON-11021566) [64]. The visual rendering was done with VMD 1.9.3 [65].

## 3. Results

At the IiWG, we sequenced 663 SARS-CoV-2 positive samples detected in Uruguay between November 2020 and April 2021 (Appendix A and Appendix A), which were classified in the following lineages: 333 (50%) P.1, 180 (27%) B.1.1.28, 70 (11%) P.2, 32 (5%) P.7, and 39 (6%) other B.1-derived lineages. The mutational profile of B.1.1.28 sequences identified 174 (26%) genomes carrying amino acid changes S:Q675H and S:Q677H that compose a new Pango lineage designated as P.6 [27,45]. Lineage P.6 was widely spread throughout the country, being detected in 12 out of 19 Uruguayan departments from 2 December 2020 to 26 April 2021; but was particularly prevalent in Montevideo, the capital department of Uruguay, and the surrounding metropolitan area (Canelones) (Figure 1A,B, right panel). Lineages P.2 and P.7, by contrast, were more frequently detected outside the metropolitan region (Figure 1B, right panel). Analysis of these 663 SARS-CoV-2 Uruguayan sequences determined by the IiWG, plus additional Uruguayan sequences obtained at the EpiCoV database in GISAID [48], revealed a changing molecular epidemiological pattern over time (Figure 1B, to the left). Lineage P.7 was the most prevalent variant (55%) detected from late November to late December 2020, but was rapidly replaced by lineage P.6, the relative frequency of which increased from 26% in December 2020 to 76% in February 2021. Lineage P.2 was first detected in Uruguay in December 2020 and reached a relatively high prevalence during the summer season, with a maximum frequency of 36% in January 2021. As previously described [26], the VOC P.1 was first detected in Uruguay in February 2021 and then quickly outcompeted the other SARS-CoV-2 variants circulating in the country, becoming the predominant lineage in April 2021.

To identify the probable geographic source of B.1.1.28, P.6, and P.7 lineages detected in Uruguay, all Uruguayan sequences here obtained (*n* = 212, Appendix A) were combined with complete genome sequences of those lineages available at the EpiCoV database in GISAID sampled in Uruguay (*n* = 143) and Brazil (*n* = 1428), and with all B.1.1.28 sequences sampled worldwide that carried mutations Q675H and Q677H (USA = 2, Spain = 1 and Belgium = 1; Appendix A). The ML phylogeographic analysis supported at least 20 independent introductions of this lineage B.1.1.28 (*n* = 14) and P.7 (*n* = 6) from Brazil into Uruguay that mostly resulted in singletons, dyads, or small clades (*n* < 5 sequences), with no evidence of extensive dissemination in Uruguay (Figure 1C and Appendix A). One B.1.1.28 introduction, however, was successfully established and originated the lineage P.6 (SH-aLRT = 98) that comprised all Uruguayan sequences (*n* = 314), as well as three sequences collected in the USA and Spain carrying mutations S:Q675H + Q677H. According to our analysis, the lineage P.6 was most likely introduced from the southeastern Brazilian region (ACR-location marginal probability (LMP) = 0.99) and was disseminated from Uruguay to the USA (two independent times, ACR-LMP ≥ 0.99) and Spain (ACR-LMP = 0.99) (Figure 1C and Appendix A). Of note, the B.1.1.28 + Q675H + Q677H collected in Belgium did not belong to clade P.6, indicating an independent and recurrent appearance of both mutations (see below; Figure 1C). We also identified one successful introduction of lineage P.7, probably from Southern Brazil (ACR-LMP = 0.99), that originated the highly supported (SH-aLRT = 100) Uruguayan clade designated as UY_P.7_ composed by 27 Uruguayan sequences plus eight sequences from Southern Brazil (Figure 1C and Appendix A). We also performed a ML phylogeographic analysis of lineage P.2 sequences sampled in Uruguay (*n* = 79) and Brazil (*n* = 1267), and identified three successful introductions, most likely from Southern Brazil (ACR-LMP > 0.85), that originated the highly supported (SH-aLRT > 0.92) Uruguayan clades UY-I_P.2_ (*n* = 35), UY-II_P.2_ (*n* = 30), and UY-III_P.2_ (*n* = 12) (Appendix A). It was noteworthy that most sequences branching within Uruguayan clades UY_P.7_ (89%), UY-I_P.2_ (100%) and UY-II_P.2_ (83%) were sampled outside the metropolitan region.

To better understand the origin and spread of lineage P.6, we performed a Bayesian phylogeographic analysis of all P.6 Uruguayan sequences produced in this work that had a geographic source available (*n* = 165), and six Brazilian basal sequences. The spatiotemporal reconstruction suggested that an ancestral B.1.1.28 virus was probably introduced into Uruguay from Brazil around October 2020, and after a short period of local evolution, the lineage P.6 ancestor arose in Montevideo (posterior state probability (*PSP*) = 0.94) around 9 November 2020 (95% HPD: 20 October–26 November) (Figure 2A). Lineage P.6 was next disseminated from Montevideo to the surrounding metropolitan area and also to more distant Uruguayan departments. The T_MRCA_ of major transmission clusters outside the metropolitan region was traced to 23 December 2020 (95% HPD: 12 December 2020 to 30 December 2020) in Rocha and 30 December 2020 (95% HPD: 19 December 2020 to 9 January 2021) in Salto (Figure 2A,B). The introduction and dispersion of lineage P.6 in each department coincided with the increase in new COVID-19 cases reported daily (Figure 2A). Given that VOI P.2 was also cocirculating, we reinforce that there was a low spatiotemporal overlap of P.6 and P.2 in Montevideo and Salto, though this was not the case for Rocha (Appendix A). The lineage P.6 was characterized by eight lineage-defining genetic changes in addition to S:Q675H and S:Q677H, including a total of five nonsynonymous mutations (Figure 2B). Of note, eight out of 10 lineage P.6-defining mutations (including S:Q675H) were also identified in a basal B.1.1.28 sequence sampled in Rio de Janeiro in January 2021, and were thus probably present in the ancestral virus that arrived from Brazil; while the remaining two mutations (ORF1ab: C8980T and S:Q677H) were fixed during the early local transmission in Uruguay. We also identified one additional mutation (ORF3a:M260I) that was fixed at a later step during evolution of P.6 in Uruguay (Figure 2B).

Structural analysis of the SARS-CoV-2 Spike glycoprotein showed that residues Q675H and Q677H were within the subdomain SD2 of each protomer constituting the homotrimer (Figure 2C). Particularly, they were located at the beginning of a very flexible loop (residues 675–690) [66], which embraced the solvent-accessible polybasic furin cleavage site [66]. These mutations were close to two experimentally observed *O*-glycosylation sites at T676 and T678 [67,68], and at the same domain of the D614G mutation [69] (Figure 2C, inset). Substitution S:Q677H has been reported as a recurrent mutation arising independently in many SARS-CoV-2 lineages, including several VOIs, circulating worldwide by the end of 2020 [70,71]. A search in the EpiCoV database (accessed on 7 July 2021) for high-quality SARS-CoV-2 genomes carrying both mutations S:Q675H and S:Q677H recovered 85 non-Uruguayan sequences. Overall, the pair S:Q675H + Q677H appeared to be distributed in 12 different countries (in decreasing frequency order: Uruguay, England, USA, Belgium, India, Australia, Switzerland, Spain, Netherlands, Japan, Germany, and France) and in 13 different Pango lineages (in decreasing frequency order: P.6, B.1.36, B.1.2, C.36, B.1.538, B.1.1.316, B.1.526 (VOI Iota), B.1.525 (VOI Eta), B.1.243, B.1.1.70, B.1.1.7 (VOC Alpha), B.1.1.63, and B.1) (Appendix A).

## 4. Discussion

Since March 2020, Uruguay had been successful at keeping the COVID-19 pandemic in check. Closed international borders and an aggressive contact-tracing system, among other government measures, were able to avoid virus transmission growing exponentially [19,20]. Brazil has been a COVID-19 hotspot in South America, and the 1068 km long Uruguayan–Brazilian dry border allowed the rapid local establishment of SARS-CoV-2 Brazilian lineages B.1.1.28 and B.1.1.33, initially associated with a few outbreaks that occurred in the departments bordering Brazil [22]. However, by the end of 2020, the pandemics worsened, with a clear increase of daily cases in December that extended until mid-February 2021. Summer-related social gatherings and relaxed social-distancing measures were some of the proposed reasons to explain the epidemic growth [20]. In this study, we described a new B.1.1.28 sublineage, designated P.6, that probably arose in Montevideo by November 2020 and spread throughout the country. Lineage P.6 comprised most (60%) Uruguayan virus genomes recovered between November 2020 and February 2021, and its spread coincided with national and local (as shown for Montevideo, Rocha, and Salto departments) increases in daily SARS-CoV-2 cases during the first epidemic wave.

Dispersion of lineage P.6 in Uruguay could have been fueled by changes in human behavior coinciding with the end of the austral spring and the ambiance of relaxed restrictions, as was demonstrated for variant 20E(EU1), which emerged in Spain and spread through Europe in the boreal summer of 2020 [72]. This hypothesis, however, failed to explain why lineage P.6 outcompeted both lineage P.7, which was the most prevalent variant in Uruguay in late 2020, and lineage P.2, which became the dominant SARS-CoV-2 strain in many Brazilian states by the end of 2020 [5,13]. An alternative hypothesis is that lineage P.6 was successfully spread in Uruguay because it was initially established in the capital city of Montevideo, which comprises nearly half of Uruguayan inhabitants and is strongly connected with all other departments. By contrast, lineages P.2 and P.7 were probably initially established outside the metropolitan region, and this may have reduced their chance of spreading at the country level. Finally, we hypothesized that the combined presence of amino acid changes S:Q675H + Q677H might have also produced a more transmissible P.6 variant, contributing to the rapid increase in the lineage dominance observed between December 2020 and February 2021.

We are not aware of any experimental assay that assessed the effect of mutations S:Q675H + Q677H on the viral fitness, but independent data indicated that these amino acid changes (either one or both of them) might facilitate viral transmissibility. These mutations are in close proximity to the polybasic cleavage site at the S1/S2 boundary that can be processed by furin and other proteases like TMPRSS2, mediating efficient entry into cells and increasing human-to-human transmission [73,74,75]. Mutations Q675H and Q677H might alter the properties of this nearby protease-cleavage site through changes in the structure conformation, glycosylation, and/or phosphorylation processes already known to have a role in cleavage regulation [73]. Recently, the promotion of syncytium formation and virus infectivity has been shown for the D614G mutation, which is known to impact structural and thermodynamic aspects of the Spike [69,76,77,78,79,80] and to enhance the protease cleavage, likely by allosterically increasing the binding to furin [81,82,83]. Knowing that histidine residues function as pH sensors in other viruses [84], Q675H and Q677H mutations might also provide some synergic structural changes in the dynamics of the subdomain SD2, enhancing the effects of mutation D614G.

Convergent evolution is a hallmark of positive selection, and we identified the independent appearance of both S:Q675H and S:Q677H in 12 additional SARS-CoV-2 lineages. Moreover, mutations close to or at the polybasic cleavage site at the S1/S2 boundary have been reported in several VOCs and VOIs, including: Alpha (S:P681H), Beta (A701V), Delta (P681R), Eta (Q677H), Iota (A701V), Kappa (P681R), and Theta (P.3, P681H). These findings suggest that the S1/S2 boundary is a region particularly relevant for selection of mutations that resulted in more transmissible SARS-CoV-2 variants. Consistent with this notion, a recent study that used a reverse genetic system and primary human airway cultures identified mutation S:P681R as a significant determinant for enhanced viral replication fitness of the VOC Delta, and supported that Spike mutations that potentially affect furin cleavage efficiency must be closely monitored for future variant surveillance [85]. Notably, another study that developed an innovative model on epidemiological variables integrating the effect of Spike amino acid changes in viral fitness forecasted that mutations Q675H and Q677H could appear in emerging SARS-CoV-2 VOCs in the following months [86]. These observations underscore the importance of future experimental studies to assess the functional impact of Spike mutations Q675H and Q677H on virus infectivity and transmissibility.

In summary, this study described the emergence and local spread of lineage P.6, a new B.1.1.28-derived lineage carrying Spike mutations Q675H + Q677H, in Uruguay that coincided with the first exponential growth phase of the country’s COVID-19 epidemic, which started by November 2020 and lasted until mid-February 2021. P.6 was the second recognized B.1.1.28-descendent lineage, together with lineage P.3, that emerged outside Brazil. An ancestral B.1.1.28 virus carrying mutation Q675H was probably introduced from southeastern Brazil into Montevideo, Uruguay’s capital city, and by November 2020 the virus already fixed mutation Q677H and spread across the entire country, originating lineage P.6. We propose that simultaneous presence of Spike mutations Q675H and Q677H might confer to lineage P.6 a higher infectivity and increased transmissibility, which, combined with the establishment in the populated metropolitan region, contributed to its swift dissemination in Uruguay. Although the lineage P.6 was substituted by the VOC P.1 as the most prevalent lineage in Uruguay since April 2021, the concurrent emergence of Spike mutations Q675H and Q677H in VOIs and/or VOCs circulating worldwide should be closely monitored.

## Figures and Tables

**Figure 1 viruses-13-01801-f001:**
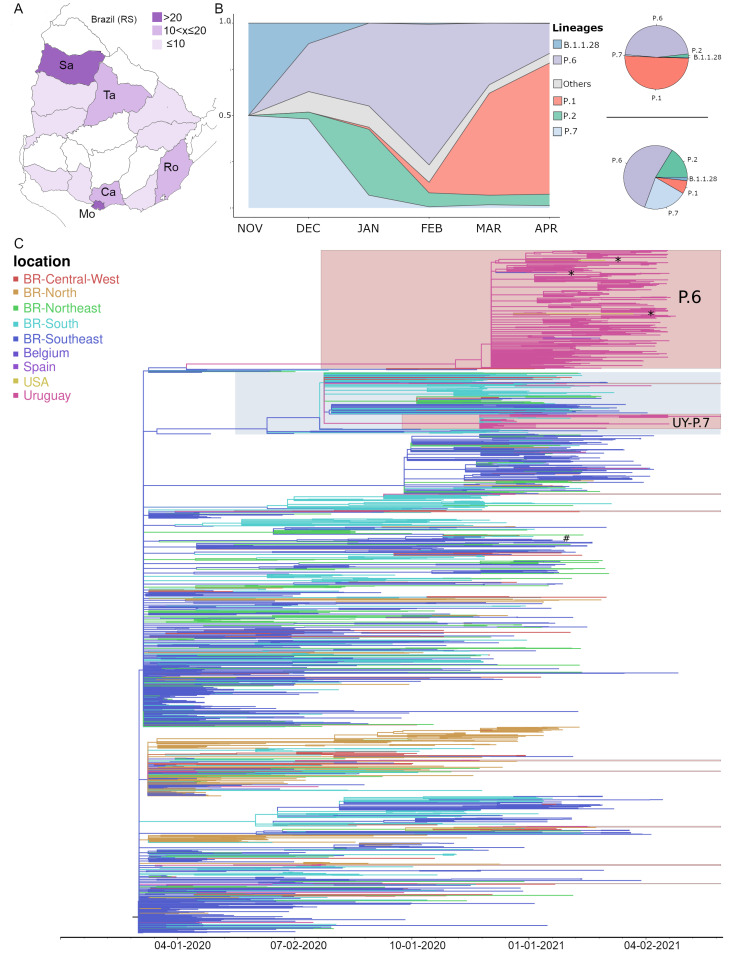
Geographic distribution, prevalence, and maximum likelihood analysis of P.6. (**A**) Map of Uruguay showing the number of sequences classified as P.6 (B.1.1.28 + Q675H + Q677H) in every department (*n* = 165 for samples with known geographical source in Uruguay). Department labels are as follows: Sa (Salto), Ta (Tacuarembo), Ro (Rocha), Ca (Canelones), Mo (Montevideo). The metropolitan area corresponds to Mo and Ca. The border with Brazil is shown. RS stands for Rio Grande do Sul, the southernmost Brazilian state. (**B**) To the left, Pango lineage proportions of all available Uruguayan samples calculated monthly, from November 2020 to April 2021. The “Others” category includes B.1, B.1.1, B.1.1.1, B.1.1.34, B.1.177, B.1.177.12, and B.1.238. To the right, pie charts show Pango lineage prevalence in the metropolitan area (top) versus the rest of the country (bottom). (**C**) Maximum likelihood phylogeographic analysis of lineage B.1.1.28 samples (*n* = 1787) from Uruguay (*n* = 355); Brazil (*n* = 1428); and the USA, Spain, and Belgium (*n* = 4) inferred by an ancestral character reconstruction method implemented in PastML. Tips and branches are colored according to sampling location and the most probable location state of their descendent nodes, respectively, as indicated in the legend. Shaded boxes highlight the major B.1.1.28 clades in Uruguay. P.6 is the assigned Pango name for the clade B.1.1.28 + Q675H + Q677H discussed here, while UY-P.7 (corresponding to UY_P7_ in the main text) is a B.1.1.28 clade carrying mutation N:P13L widely distributed in southern Brazil (recently assigned as P.7). Brazilian P.7 is shown with a gray shadow. Asterisks (*) indicate the sequences dispersed from Uruguay to the USA and Spain. The hash indicates the B.1.1.28 + Q675H + Q677H sample from Belgium. The time-scaled tree was rooted with the earliest sequence (collection date 5 March 2020). Branch lengths are drawn to scale indicating nucleotide substitutions per site per year.

**Figure 2 viruses-13-01801-f002:**
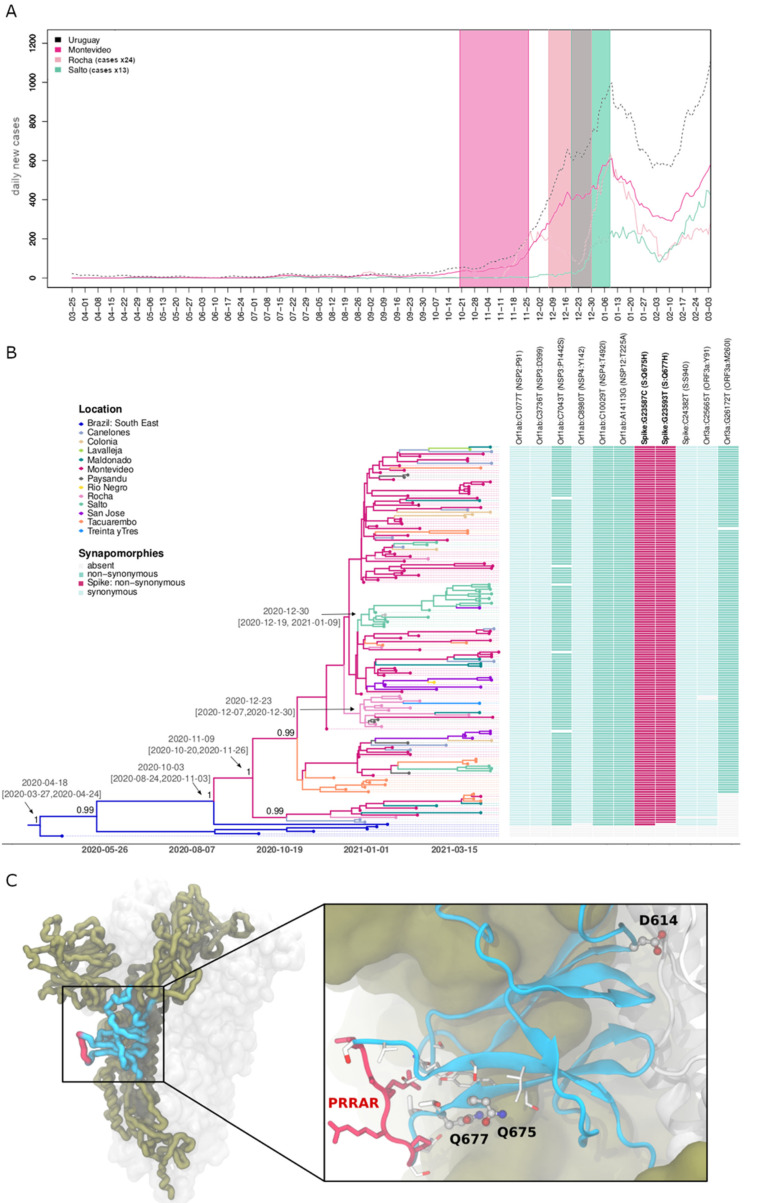
Bayesian phylogeographic analysis and description of synapomorphies of P.6. (**A**) Number of daily new cases from March 2020 to March 2021 in the country (black), in Montevideo (fuchsia), Rocha (pink), and Salto (green). Daily new cases for Rocha and Salto were multiplied by a factor proportional to the population of that department in comparison to Montevideo (times 24 and 13 for Rocha and Salto, respectively) for visualization purposes. Confidence intervals of TMRCA of Montevideo (fuchsia), Rocha (pink), and Salto (green) P.6 clades are shown as shaded areas. (**B**) Bayesian phylogeographic analysis of the P.6 clade in Uruguay, implemented in BEAST. Uruguayan sequences generated by the IiWG with known geographic source (*n* = 165) were combined with six additional basal sequences from southeastern Brazil. Tips and branches of the time-scaled Bayesian tree are colored according to sampling location and the most probable location state of their descendent nodes, respectively, as indicated in the legend. Posterior probability support values and estimated TMRCAs are indicated at key nodes. Additionally, a heatmap represents the presence or absence of 10 synapomorphic sites and an additional change (ORF3a:M260I) shared by most sequences. The color scheme indicates the different mutations, as indicated in the legend. In each case, genomic position, nucleotide substitution, viral protein, and amino acid are shown. (**C**) One Spike protomer of the homotrimer is shown in thick ribbons, while the others are represented as transparent solvent accessible surfaces. The subdomain SD2 (residues 590–700) is indicated in blue, and the polybasic furin cleavage site (PRRAR, residues 681–685) is in red. The inset shows a zoom into the structural context of SD2, representing residues D614, Q675, and Q677 in balls and sticks, and nearby residues in sticks.

## Data Availability

All SARS-CoV-2 genome sequences have been submitted to the EpiCoV/GISAID database with accession numbers indicated in Appendix A.

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
