# Peer review of "Emergence and Spread of a B.1.1.28-Derived P.6 Lineage with Q675H and Q677H Spike Mutations in Uruguay"

_viruses, 2021, doi:10.3390/v13091801_

Round 1

Reviewer 1 Report

The study by Rego et al describes the emergence of the B.1.1.28 lineage in Uruguay in 2020. The lineage is characterized by two amino acid substitutions in the spike protein, which might have increased viral fitness. However, this lineage was subsequently replaced by P.1 in April 2021. I enjoyed reading the study, the methods are sound, and the results well presented. It’s important for individual countries to understand their transmission history and which factors most contribute to the emergence and dissemination of clusters. I absolutely loved the pastel colour palette used in the figures! However, I think they could be improved even more and I have some other suggestions.

Minor comments.

  1. What is the proportion of sequences used in the study compared to the number of cases? Apologies if I missed it in the text.

  1. The authors mention the border to Brazil as being crucial in the ‘importation’ of variants. However, the numbers in this study don’t seem to corroborate that. According to this study, most sequence data is from South and West.

Figures general

  1. Increase font size. Most labels are too small.

Figure 1

  1. Some labeling within the map would be helpful. I suggested adding the locations for Montevideo, Rocha, and Salto, as well as indicate the borders to Brazil.

  1. I’d suggest increasing the size of the legend for the phylogenetic tree. The font size is rather small. Please add the branch length reference in the figure legend (I assume it’s nucleotide substitutions per site per year).

  1. I would also suggest rearranging the panels in figure one such that the tree can be increased in height. This would improve the visualization of individual clades within the tree. At the moment the tree is rather squished and the UY-N:P13L clade is not easy to spot.

Figure 2

  1. Enlarge the legend.
  2. The legend does not reflect the line style, e.g. black dotted should be marked as such and not a black square.

I would add the amplification factor to the legend in the figure too. E.g. Rocha x24

And add a label to the y axis

  1. In the heat map there are 11 columns for 11 mutations but in text reads that P6 has 10 lineage-defining mutations. What does the heat map show?

In general, I did not find the heat map very informative as these are lineage-defining mutations so there are very few differences within the tree.

Author Response

REVIEWER #1

Comments and Suggestions for Authors

The study by Rego et al describes the emergence of the B.1.1.28 lineage in Uruguay in 2020. The lineage is characterized by two amino acid substitutions in the spike protein, which might have increased viral fitness. However, this lineage was subsequently replaced by P.1 in April 2021. I enjoyed reading the study, the methods are sound, and the results well presented. It’s important for individual countries to understand their transmission history and which factors most contribute to the emergence and dissemination of clusters. I absolutely loved the pastel colour palette used in the figures! However, I think they could be improved even more and I have some other suggestions.

We are very grateful to the reviewer’s comments. We have followed all suggestions, as explained below. However, given the comments of the second reviewer, there are major changes in the manuscript. We have greatly improved the introduction, and also incorporated changes in the Methods and Results sections. The changes are related to the reorganization of a previous manuscript that we had elaborated but now dismissed (https://www.medrxiv.org/content/10.1101/2021.07.05.21259760v1). All sequences generated in that previous work have been incorporated here, thus sequences produced in the current manuscript have increased from 180 to 260. Tables with sequences data, methods and results have been changed accordingly. Additionally, we have removed all references citing this preprint and we have added three authors that were part of that work that generated the additional sequences. We have added a new supplementary Figure S1. All main results and main figures have remained the same. By including this complementary data set we have improved the quality of the present manuscript.

Minor comments.

  1. What is the proportion of sequences used in the study compared to the number of cases? Apologies if I missed it in the text.

A1: We thank the reviewer for this comment. We missed putting this information in the manuscript. We now added a supplementary figure S1A, where this information is given. Percentages of sequences generated by our group compared to the total cases are between 0.2% and 0.8%. As the IiWG was launched in March, 2021, older samples are scarce and, after P.1 introduction by mid-February, monthly COVID-19 cases skyrocketed. Even though the sequenced samples increased, the percentage of analyzed cases did not change a lot. 

  1. The authors mention the border to Brazil as being crucial in the ‘importation’ of variants. However, the numbers in this study don’t seem to corroborate that. According to this study, most sequence data is from South and West.

A2: The authors of this study, in the context of the IiWG or previous projects, have been following SARS-CoV-2 introductions in the country since the beginning of the pandemics. We have established that Brazil has been a big contributor in the SARS-CoV-2 lineages and variants circulating in Uruguay, explained mainly because there is a 1,068 km long dry border with southern Brazil which includes five twin cities. Most of these results are already published and referenced in the current manuscript (Mir et al 2021, Rego et al 2021). However, as noted by the reviewer and described in the manuscript, while previous introductions are sourced in southern Brazil, lineage P.6 is inferred to have Montevideo as its origin and its close related sequences available in Brazil belong to the southeast region (specifically Sao Paulo and Rio de Janeiro states). In fact, this inferred dissemination from Montevideo, instead of border departments, gave as a clue about a different and interesting new pattern in the epidemics which deserved further study. We hope that the modification of the map in Figure 1A also contributes to clarifying these statements. 

Figures general

  1. Increase font size. Most labels are too small.

A3: We increased the fonts of almost every figure.

Figure 1

  1. Some labeling within the map would be helpful. I suggested adding the locations for Montevideo, Rocha, and Salto, as well as indicate the borders to Brazil.

A4: As suggested, we added the border with Brazil and the locations of some of the relevant departments in Uruguay. We indicated this in the figure legend, as follows:

“Department labels are as follows: Sa (Salto), Ta (Tacuarembó), Ro (Rocha), Ca (Canelones), Mo (Montevideo).  The border with Brazil is shown. RS stands for Rio Grande do Sul, the most southern Brazilian state.”

  1. I’d suggest increasing the size of the legend for the phylogenetic tree. The font size is rather small. Please add the branch length reference in the figure legend (I assume it’s nucleotide substitutions per site per year).

A5: We increased the size of the legend in figure 1C. We added the branch length reference in the legend, as follows: 

“Branch lengths are drawn to scale with the bar indicating nucleotide substitutions per site per year.”

  1. I would also suggest rearranging the panels in figure one such that the tree can be increased in height. This would improve the visualization of individual clades within the tree. At the moment the tree is rather squished and the UY-N:P13L clade is not easy to spot.

A6. While we have kept the position of the panels in figure 1, we increased substantially the height of the tree in figure 1C. Clades are more clear and visible now.

Figure 2

  1. Enlarge the legend.

A7: As suggested, we increase all legends in figure 2.

  1. The legend does not reflect the line style, e.g. black dotted should be marked as such and not a black square.

I would add the amplification factor to the legend in the figure too. E.g. Rocha x24

And add a label to the y axis

A8: As suggested, the legend was modified and the amplification factor was added to the figure legend and the y-axis was labeled.

  1. In the heat map there are 11 columns for 11 mutations but in text reads that P6 has 10 lineage-defining mutations. What does the heat map show?

In general, I did not find the heat map very informative as these are lineage-defining mutations so there are very few differences within the tree.

A9: In the heatmap we show the 10 lineage-defining mutations together with an additional column (11th column) to show one non-synonymous mutation that is present in most P.6 sequences although is not a P.6 lineage-defining change. We are grateful to the referee because we previously planned to add the explanation in the legend but missed it. We have added a line in the main text as follows:

“This new lineage P.6 spreading in Uruguay is characterized by 10 lineage-defining genetic changes, including five non-synonymous mutations, two of them in the Spike viral protein (Figure 2B). One additional amino acid change (ORF3a:M260I) is shared by most sequences in the clade (Figure 2B).“

and also changed one sentence in Figure 2 legend:

“Additionally, a heatmap represents the presence or absence of 10 synapomorphic sites and an additional change (ORF3a:M260I) shared by most sequences. The color scheme indicates the different mutations, as indicated in the legend. In each case, genomic position, nucleotide substitution, viral protein and amino acid are shown.”

We find the heatmap rather useful to visualize especially the type of mutations that define the lineage and where those are located (e.g. we can see easily that two of the ten fall within the Spike).

Reviewer 2 Report

The manuscript entitled: “Emergence and spread of a B.1.1.28-derived lineage with Q675H 2 and Q677H Spike mutations in Uruguay” by Rego et al. seems as a large-scale study of the genetic diversity of the SARS-CoV-2 viral lineages identified in different regions of Uruguay during the pandemic period November 2020 - April 2021. In fact, this is a partial continuation of the previous studies (https://doi.org/10.1101/2021.07.05.21259760 and https://doi.org/10.3389/fmicb.2021.653986) of the same team, but here the results are interpreted in a different way. The novelty of this manuscript are the mutations (Q675H and Q677H) found in the B.1.1.28 viral lineage.

Although the manuscript presents some interesting data about the spreading of B.1.1.28-derived lineages, many critical issues, reported below, must be resolved before publication.

  1. Section “Introduction” is very short – it should be extended.
  2. The main goal of the study has to be clearly shown.
  3. The authors have to distinguish and identify what is the new in this study compared with their previous reports mentioned above. For example, here is not given the number of studied patients and whether they are the same or different patients.
  4. Discussion about the possible role of these mutations for spreading of the virus has to be extended, even more, that this lineage (B.1.1.28) during the next period has been replaced by others (VOI P.2 and the VOC P.1).

Minor points:

  • The size of the text of the figure legends needs to be increased. It is not always readable.
  • Author Contributions: The authors with similar contribution have to be combined (MA, TP, NR, MNB, AL: resources, investigation. NOT MA: resources, investigation, TP: resources, investigation, NR: resources, investigation. MNB: resources, investigation. AL: resources, investigation.

Author Response

Comments and Suggestions for Authors

The manuscript entitled: “Emergence and spread of a B.1.1.28-derived lineage with Q675H 2 and Q677H Spike mutations in Uruguay” by Rego et al. seems as a large-scale study of the genetic diversity of the SARS-CoV-2 viral lineages identified in different regions of Uruguay during the pandemic period November 2020 - April 2021. In fact, this is a partial continuation of the previous studies (https://doi.org/10.1101/2021.07.05.21259760 and https://doi.org/10.3389/fmicb.2021.653986) of the same team, but here the results are interpreted in a different way. The novelty of this manuscript are the mutations (Q675H and Q677H) found in the B.1.1.28 viral lineage.

Although the manuscript presents some interesting data about the spreading of B.1.1.28-derived lineages, many critical issues, reported below, must be resolved before publication.

We are very grateful for the reviewer's comments and suggestions. In fact, we have dismissed the preprint (https://doi.org/10.1101/2021.07.05.21259760) and reorganized the datasets accordingly. Thus, the reviewer will see a change in the number of sequences in each partition but the total numbers remain the same (e.g. some sequences are now presented in Table S1 instead of Table S2). Not only methods and results have been modified and improved, but also the introduction has been improved a lot. We believe that the manuscript has gained a lot in clarity. 

  1. Section “Introduction” is very short – it should be extended.

A1: The introduction was properly extended. In fact, the introduction now has about 800 words and 27 references.

  1. The main goal of the study has to be clearly shown.

A2: In the new introduction we better explained the goal of our study. We included following statements:

“... there is a gap in knowledge concerning the factors that fueled SARS-CoV-2 viral dynamics during the first exponential increase of COVID-19 through the end of 2020 and beginning of 2021. 

To understand the SARS-CoV-2 diversity associated with the first COVID-19 epidemic wave in Uruguay, we conducted a retrospective epidemiological and genomic analysis of SARS-CoV-2 complete genomes. In the context of the Inter-Institutional Working Group (IiWG) for SARS-CoV-2 genomic surveillance in Uruguay launched in March, 2021 (Rego et al., 2021), additional samples from COVID-19 patients diagnosed between November, 2020 and February, 2021 were recovered, revealing that lineage B.1.1.28 was the most prevalent viral lineageSARS-CoV-2 variant by the end of 2020 and beginning of 2021.”

  1. The authors have to distinguish and identify what is the new in this study compared with their previous reports mentioned above. For example, here is not given the number of studied patients and whether they are the same or different patients.

A3: As we explained above, we dismissed the preprint (10.1101/2021.07.05.21259760). We have reorganized the datasets and included all patients in the current manuscript. Thus, there are 260 patients included in the present study. While 212 obtained genome sequences correspond to B.1.1.28, 174 are within the new P.6 lineage (B.1.1.28+Q675H+Q677H). We used additional samples only for genotyping purposes or in the main ML tree. Some of them are generated by us and are available at the IiWG domain but not yet publicly available (mostly P.1 sequences). 

The rest of them are Uruguayan sequences available in GISAID generated by the New York University. These sequences have been used to calculate lineage prevalence and the B.1.1.28 sequences were incorporated in the big ML tree. However we decided not to use the New York’s P.6 sequences in the bayesian phylogeographic analyses, since these sequences have not been cited in any manuscript or pre-print yet and we preferred not to use them in those specific analyses as we feel uncomfortable regarding ethical aspects. Moreover, the sequences were collected in February, 2021 or later, so we think they can not substantially contribute to the inferences about what happened earlier during P.6 emergence and first dissemination.  

The sequences discussed in this work, compared to the national monthly COVID-19 cases, are now shown in the new Figure S1A.

  1. Discussion about the possible role of these mutations for spreading of the virus has to be extended, even more, that this lineage (B.1.1.28) during the next period has been replaced by others (VOI P.2 and the VOC P.1).

A4: We have better explained the dissemination of P.2 and P.6 in the country. P.2 did not show country-wide spread as P.6 (we added additional Figures S1B and S1C) and data does not suggest that P2. replaced P.6. However, after P.1 introduction, P.1 outcompeted P.6 and every other lineage circulating in the country. As far as we understand, data suggests a role of Q675H and Q677H in increasing viral transmissibility, maybe contributing to viral fitness already increased by D614G. We are not suggesting these mutations can improve virus fitness as can be seen in VOC P.1. We firmly believe that these mutations, situated nearby to the S1/S2 cleavage site, are in a region where many changes can modify protein conformation and/or glycosylation and phosphorylation patterns which in turn can modify cleavage regulation, an thus affect the virus capacity to infect cells. However, we do not have any experimental data, hence we have extended our discussion but kept it cautiously speculative.

Minor points:

  • The size of the text of the figure legends needs to be increased. It is not always readable.

A: We increased font size in all legends.

  • Author Contributions: The authors with similar contribution have to be combined (MA, TP, NR, MNB, AL: resources, investigation. NOT MA: resources, investigation, TP: resources, investigation, NR: resources, investigation. MNB: resources, investigation. AL: resources, investigation.

A: We added the following Author Contribution statement.

Conceptualization, N.R., G.B., L.S.; methodology, C.S., M.P., A.C., A.F., P.P., V.N.; software, I.F.; formal analysis, N.R., C.S., ,M.P., A.C., A.F. , P.P, M.R.M, G.B..; investigation,  N.R., C.S., M.P., A.C., A.F., P.P, T.F., V.N., M.B., R.A., M.A., T.P., N.Re., M.N.B, A.L., V.B., A.M., O.C., N.N., J.H., M.D., B.G., L.G., M.Me, M.P.T, J.Z., B.R., M.M., M.A., P.S., D.M., C.A., J.M., H.A., R.C., G.B., P.M., G.M., G.I., L.S.; resources, V.N., M.B., R.A., M.A., T.P., N.Re., M.N.B, A.L., V.B., A.M., O.C., N.N., J.H., M.D., B.G., L.G., M.Me, M.P.T, J.Z., B.R., M.M., M.A., P.S., C.A., J.M., H.A., R.C..; data curation, N.R., C.S., I.F., T.F..; writing—original draft preparation, N.R., L.S; writing—review and editing, C.S., T.F., M.R.M, D.M., G.B., P.M., G.M., G.I.; visualization, M.R.M.; supervision, P.M., G.M., G.I., L.S.; funding acquisition, P.M, G.M, G.I. 

Round 2

Reviewer 2 Report

The manuscript has been significantly improved.